# An Assessment of Chinese Pathways to Implement the UN Sustainable Development Goal-11 (SDG-11)—A Case Study of the Yangtze River Delta Urban Agglomeration

**DOI:** 10.3390/ijerph16132288

**Published:** 2019-06-28

**Authors:** Xueyan Xu, Jun Gao, Zhonghao Zhang, Jing Fu

**Affiliations:** 1Department of Tourism Management, Shanghai Business School, Shanghai 200235, China; 2Institute of Urban Studies, School of Geography and Environmental Science, Shanghai Normal University, Shanghai 200234, China; 3Northwest Institute of Eco-Environment and Resources, Chinese Academy of Sciences, Lanzhou 73000, China

**Keywords:** urban sustainability assessment, sustainable development goals, full permutation polygon synthetic indicator (FPPSI) method, Yangtze River Delta urban agglomeration

## Abstract

Urban sustainability is a crucial part of the United Nations (UN) Sustainable Development Goals (SDGs) and one of the core objectives of China’s national strategy to promote new urbanization and achieve integration in the Yangtze River Delta (YRD). This paper mainly focused on the 11th SDG, which is a universal call to make cities and human settlements inclusive, safe, resilient and sustainable. The full permutation polygon synthetic indicator (FPPSI) method was applied to synthetically evaluate the sustainable level of 26 cities in the YRD urban agglomeration from 2007 to 2016. The results showed that: (1) the synthesis indicators were increasing year by year, which implied that the sustainable development of the YRD has shown obvious progress in recent years. However, each city faced its own challenges to achieving the sustainable development goals. The sustainability level for the majority of cities was restricted by obstacles such as the per capita green area, air quality and commercial housing sales area; (2) Among the 26 cities, small and medium-sized cities were subject to the traditional strong sustainability indicators while large and mega cities were more affected by weak sustainability indicators; (3) Spatial differences were found for the overall sustainable development level of the YRD. The diffusion and assembly effect among cities had not yet been formed; however, the strong spillover effect of developed cities might influence the ability of other cities to achieve sustainable development goals in many aspects of the environment, economy and society. The results suggest the need for a stronger focus on improving regional developing patterns and strengthening coordination in the process of achieving the sustainable development goal of urban agglomeration in the YRD. Furthermore, according to the conditions of different cities, integrated policies are required to address all aspects of sustainability and to avoid unintended consequences.

## 1. Introduction

In September 2015, the United Nations (UN) adopted a list of SDGs (Sustainable Development Goals) to aid in the creation of framework arrangements for the three aspects of global economic development, social equity and ecological protection across the period of 2015–2030. Specifically, the SDGs are 17 universal goals that meet the urgent environmental, political and economic challenges facing our world (United Nations 2015). The document provides a framework for global sustainable development. SDGs are mainly concentrated on cities, energy, health, climate and poverty eradication. The latter is regarded as a key component of MDGs (Millennium Development Goals), emphasizing sustainable and inclusive development in multiple dimensions with a higher degree of universality and flexibility. Based on the SDG-13 (Climate action), Raheem counseled urgent action throughout the world to address climate change and its impacts. Taking Africa as an example, regional differences and diversified development needs are the main obstacles in achieving this goal [1]. At the same time, poverty eradication is also the main challenge facing the world [2]. Bryan et al. scientifically examined China’s SDGs practices and found that the results of 16 major projects aiming at improving the sustainable development of the environment and people’s livelihood were very positive but the country’s overall sustainable development was also influenced by population policies [3]. Compared with the previous various sustainable evaluation index systems, the SDG indicator system is based on global sustainable evaluation, which makes the results more comparable among different regions with higher credibility. The UN Sustainable Development Goals are a large and complex evaluation system. In order to achieve the goals throughout the world, most of the goals have been described in a vague way. It is thus difficult to have unified measurement standards and data sources. The problem is that we need to understand international benchmarking in the future, measuring the planning and construction of cities with scientific and diversified standards in order to explore the sustainable development pathways for the future [4] Despite the broad literature on issues related to sustainability, there is still a lack of studies aiming at investigating this subject deeply through the description and assessment of the comprehensive sustainable development index [5].

More than half of the world’s population now live in urban areas, which means urban has become a major component of human civilization as well as a key concern for sustainable development [6]. Therefore, studies on urban sustainable development assessment account for a large number of the existing papers on SDGs. The future of SDG-11 is expected to include opportunities for everyone and access to basic services, energy, housing and transportation. It focuses on measuring the elements of sustainable urban development, including housing and slum upgrading, sustainable transportation systems, air quality improvement, public green space and so forth [7,8,9]. In view of the land encroachment on rural areas caused by rapid urbanization, tight resources, increased pollution and forced transformation of the urban economy, SDGs (Sustainable cities and communities) also pay attention to safety, tolerance, resilience and sustainability in urban development. Scholars generally believe that the inclusion of cities in SDGs is extremely challenging because urban development involves many stakeholders and is highly prone to regional imbalances. The case studies in Bangalore, Cape Town, Gothenburg, Manchester and Kisumu found that each city had data missing varying extents [10]. Raheem took Nigeria as an example to investigate the impact of the spatial agglomeration of urban resource scarcity on the health status of Ilorin residents [1]. Cai et al. explored the relationship between SDG-3 (Good health and well-being) and -11 (Sustainable cities and communities). The results showed that blindly focusing on global health and health indicators caused negative impacts on urban development goals [11,12]. Arslan et al. (2016) used the indicators in SDG-11 to evaluate the sustainability of energy and environmental planning in three ancient cities [13]. In addition, Gao et al. found that the UN Sustainable Development Goals are difficult to implement collectively, and they also sorted out many problems in evaluating sustainable development such as the applicability and logical rationality of sustainability indicators [14]. However, almost all these studies focused merely on the effects and impacts of single indicators.

To date, research on urban sustainability is still in the early stages; especially compared with the qualitative exploration research, the quantitative empirical research is quite limited [15,16]. In addition, the principles of the most commonly methods used in sustainability performance evaluation are the entropy method, the grey approach, the analytic hierarchy process (AHP) and the equal weighting method [17,18]. Previous studies also showed that the assessment of sustainability is often selected from different subsystems and criteria. Therefore, researchers started to employ the full permutation polygon synthetic indicator (FPPSI) method, which can reflect the integrative system principle by altering the traditional additive approach to combine indicators in the quantitative evaluation of urban sustainability [19,20].

Since its reform and opening up in 1978, China has been faced with the overlapping and interweaving of urban and rural space as well as the high concentration of population and economy. The solid erosion, the heat island effect, air, water and solid waste pollution result into greater environmental pressures and economic loss in the process of rapid urbanization [21,22,23,24]. Such great threats and risks have brought about a crisis of unsustainable development for the city [25,26,27,28]. Urban agglomerations are crucial developing patterns for the future of China. In addition, the development of and competition among modern cities no longer relies on the expansion and strength of a single city but the growth of urban agglomeration. As a typical economic and population-intensive area of China, the Yangtze River Delta (YRD) also faces prominent problems such as accelerated expansion and increased pollution [29]. Nowadays, the YRD region plays an important role in the national strategy for establishing Chinese pathways to implement SDGs. How to coordinate the relationship between land development and utilization, in addition to strengthening environmental protection and industrial innovation, is the current issue [30]. Based on the framework of SDG-11, this paper aims to establish a set of substitution indicators that can summarize the key characteristics of each city in the YRD for a long-term and systematic search. This study also tried to scientifically, systematically and rigorously explain the status and characteristics of sustainable development in Chinese cities and give recommendations on the applicability and path of implementation for the UN’s Sustainable Development Goals.

## 2. Study Area

The YRD is one of the regions with the most diversity, openness, innovation and the absorption of the largest numbers of immigrants in China. The study area comprises 26 cities, including Shanghai, Nanjing, Wuxi, Changzhou, Suzhou, Nantong, Yancheng, Yangzhou, Zhenjiang, Taizhou, Hangzhou, Ningbo, Jiaxing, Huzhou, Shaoxing, Jinhua, Zhoushan, Taizhou, Hefei, Wuhu, Maanshan, Tongling, Anqing, Zhangzhou, Chizhou and Xuancheng (Figure 1), with a total land area of 217,700 square kilometers, a GDP of 14.72 trillion yuan in 2016 and a total population of more than 150 million people, accounting for 2.2%, 19.78% and 11.0% of the whole country respectively.

With the rapid economic and social development of the YRD, there are potential problems, such as the increasing prominence of urban issues like energy consumption pressure, cultivated land loss, water shortage, serious environmental pollution, aging population and increasing population burden [31,32]. These issues bring ongoing challenges to the sustainable development of urban agglomerations. In 2018, the integration of the YRD was identified as a national strategy to improve the spatial distribution of China’s reform and the opening up of China. Over the next three decades, the YRD will make great infrastructure investments and formulate joint policies to relieve the pressure on the core areas and increase productivity on a national basis. Getting the planning and process of the YRD agglomeration right is key to building ongoing sustainable growth and establishing an example of how to make China’s cities productive, livable and environmentally friendly by 2050 (The State Council of the People’s Republic of China 2018).

## 3. Methods and Data

### 3.1. Indicators Related to SDG-11

Referencing to the preceding interpretation of SDGs-11 and the accessibility of research data, a total of 11 indicators were selected to construct the index system in six themes: Housing, traffic, land use and participatory planning, environmental impact, public space and relationship between urban and rural areas. With respect to Urban sustainability science, the ‘‘weak’’ and ‘‘strong’’ substitutability indicators can comprise environmental, political and economic challenges in achieving SDGs. What’s more, the strong sustainability at a broad scale may not be achieved without a proper combination of weak sustainability on smaller scales, more attention should be paid to emphasize the difference between artificial and natural attributes of the capital resources, we developed and grouped 11 urban-related SDG-11 indicators into two categories—weak and strong sustainability indicators [33,34]. 

The final list of 11 indicators considered in the present analysis is reported in Table 1. The first column is the target layer connects each indicator to UN SDG-11. The second column reports the code name used in the result section (Section 4). The last column shows the attribute of capital resources by divided the indicators into week and strong sustainability factor. As shown in Table 1, the strong sustainability indicators of the analysis include urban sewage discharge, solid waste discharge, domestic garbage disposal rate, PM_2.5_ annual average concentration and days of good air quality. Meanwhile, the weak indicators include commercial housing sales area, buses, urbanization rate, urban green space area, per capita park and green space area and household registered population. The data were collected from the China City Statistical Yearbook and the National Bureau of Statistics of China (2007–2016). In order to achieve vertical comparability of the indicator system, the indicators are processed in a positive and negative manner. When the index has a “+” indicator, the datapoint is taken as a positive value and vice versa. 

### 3.2. The Full Permutation Polygon Synthetic Indicator Method

The full permutation polygon synthetic indicator method was used for data standardization and indicator synthesis [33]. It included rescaling each indicator on a scale from −1 to 1 and then altering the traditional additive approach to combine indicators by using a multidimensional approach which could better reflect the integrative system principle. This advantage enables the sensitive identification of cities with upper and lower values for each indicator and highlights hot spots and cold spots visibly. In order to quantitively evaluate the sustainable level in these 26 cities of the YRD, each city can be compared at various times and in all dimensions to achieve the calculation of each value and sustainable level of the cities. An n-sided polygon is created to represent each of the n indicators and the upper limit of these indices is a radius forming a central positive n-gon and the connecting lines of the index values constitute an irregular center n-gon. The vertices of this irregular center n-gon are a full permutation polygon of n indicators from end to end. Thus, the n indicators can form a total of (*n*−1)! /2 different irregular center n-gons, which can be used to evaluate each objective function:(1)$$F(x)=\frac{a(x+b)}{x+c},a\neq 0,x\geq 0,$$

*F*(*x*) satisfies *F*(*L*) = −1, *F*(*T*) = 0, *F*(*U*) = 1, where *L* is the lower limit of the index *x*, *U* is the upper limit of the index *x* and *T* is the threshold of the indicator *x*. According to the above three conditions, we can obtain
(2)$$F(x)=\frac{\left(U-L)(x-T\right)}{(U+L-2T)x+UT+LT-2UL},x\in [L,U].$$

The standardized calculation formula for calculating the *i*-th indicator is
(3)$$S_{i}=\frac{(U_{i}-L_{i})(x_{i}-T_{i})x}{(U_{i}+L_{i}-2T_{i})x+U_{i}T_{i}+L_{i}T_{i}-2U_{i}L_{i}}.$$

The formula for calculating the composite index based on fully arranged n-gons is
(4)$$S=\frac{1}{2n(n-1)}\sum_{i=1}^{n}\sum_{j=1}^{n}\left[(S_{i}+1)][(S_{j}+1)\right],i\neq j.$$

In Formula (4), *S_i_* and *S_j_* are the indicators for *i* and *j* and *S* is the value of the synthetic indicator for each city. The value of *S* is between [0, 1] and larger value that is considered, the better. *F*(*x*) takes a value between [−1, 1] and the theoretically largest value is the farthest from the center of the n-gon, which is normalized to have a value set to 1. In contrast, the center of then-gon has a value of −1. Since setting different weights for the indicators of each type of city might bring about great uncertainty and end in results that are not comparable, all the related indicators were all equally weighted [20,34].

## 4. Results and Discussion

### 4.1. Levels of the Urban Sustainability Indicators

The empirical analysis of this paper includes an evaluation of the sustainable development performance in the YRD from 2010 to 2016, which examined the changes in the related indicators and synthesis indicators of SDG-11 before and after the launch of the UN Sustainable Development Goals. The results show that the synthesis indicators in the YRD increased year by year. The urban planning tended to be reasonable and several urban-related SDG-11 indicators fluctuated slightly in different years. To provide a preliminary indication of potential trajectories for the next few years, we assessed each urban-related SDG-11 indicator from 2007–2016 (Figure 2).

Concerning city size, the urbanization level of the YRD municipality and the provincial capital city had slowed down quite markedly. The main growth areas of the growth point included Shanghai, which grew with an urbanization rate of over 80%; Jiangsu, which grew by 55.6%; and Zhejiang, which grew by 57.9%. The urban agglomeration network system centered on Shanghai, Nanjing and Hangzhou, as sub-level centers had been initially formed. In the past 10years, the number of registered households in Anhui Province dropped significantly with a large number of people flowing out; in contrast, other cities were relatively stable. As the mega city, Shanghai was the only one with a registered population of over 10 million. The growth of housing sales area was still concentrated in Shanghai, Hangzhou, Hefei and other municipalities directly under the central government and provincial capitals. Only the housing sales area of Suzhou was larger than that of the provincial capital city, Nanjing, as it attracted many investments and achieved great development in the last decades.

As for the urban environmental impact, the overall urban sewage discharged from major cities in the YRD showed a steady and small upward trend in the past decade. The discharge of solid waste experienced a process of increasing at first, then decreasing and then increasing again throughout the period of 2007 to 2016. The cities that contributed a lot to the emission indicators in the early period were Shanghai, Suzhou and Maanshan. In recent years, since the standardized management systems and standards for various types of waste have been gradually implemented, the values of emission indicators have declined. The domestic garbage disposal rate has reached 100% for most cities since 2010. The average annual air quality was maintained in good condition between 2007 and 2009. Because of the introduction and implementation of a more severe national air quality standard, a significant deterioration took place after 2010 but the air quality gradually returned to good condition over the last two years. The air pollution in Hefei, Anqing, Chizhou, Yangzhou and Tongling is still relatively severe. Zhoushan, which has good coastal diffusion conditions and low industrialization, was the only city that had performed well all year round in terms of the maintenance of air quality. In general, industrial structure has an important impact on environmental pollution. With the continuous strengthening of environmental policies in the YRD, environmental pollution was greatly relieved, and the conditions of the coastal areas were found to be in relatively better conditions than those inland.

The area of urban green space in the YRD increased slightly in the past decade. Due to the population density, the per capita park and green space area of Shanghai was at the bottom of all the cities in the YRD. In recent years in the YRD, due to the strict control of urban development boundaries and the optimization of urban and rural spatial structures, the central government implemented a new greening action plan by building a number of urban parks and national parks. However, the high density of the urban population still causes tensions in public space resources.

### 4.2. Synthesis Indicators from 2007 to 2016

The synthetic indicator for the sustainability of 26 cities from 2007–2016 exhibited a descending trend (Figure 3). The pioneer cities were Hangzhou, Ningbo, Shanghai, Nanjing and Zhoushan, while Maanshan, Taizhou, Tongling, Anqing and Xuancheng dominated the lowest of the urban-related SDG-11 index. The positive rate of change in Yancheng, Taizhou and Shanghai was high; to the contrary, the rate of negative change in Chizhou and Tongling was high. 

Thus, there existed a polarized phenomenon in the sustainable development in the YRD agglomeration. The sustainability level for those cities was divided into four types (levels I–IV).According to which level the highest score belonged to, some of the credit for the excellent performance of cities (level I) like Hangzhou, Shanghai, Suzhou and Nanjing must go to their coastal location, developed transportation network and high level of public services. As representatives of level II, Wuxi, Zhoushan, Hefei and Changzhou had one or two indicators perform badly and the others performing well or excellently. Meanwhile, the synthetic scores of those cities experienced the trend of rapid development and obvious decline during the last five years of 2007 to 2016. This indicated the emergence of hidden dangers from rapid development and the rise of competitors among urban agglomerations. Moreover, the air pollution problems generated in the short term caused those cities to perform poorly in the sustainability index. The level III cities exhibited so-called neutral development, with almost no indicators performing excellently or very badly. In terms of time span, cities like Zhenjiang, Chizhou, Shaoxing and Xuancheng, which are rich in natural resources but lacking in technological development, faced a lot of pressure in recent decades. The increasing industrialization level of cities could also aggravate the risk and instability of sustainable development. To be specific, those cities suffered from a low quality of life, which was caused by the rapid expansion of the city’s size and a large number of emigrants. In the future they can try to maintain a good level of development through technology innovation, the introduction of new talent and the construction of infrastructure. Cities (level IV) represented by Maanshan and Anqing—with several scores less than 0 and some reaching −1—were representative of those cities that performed poorly or very poorly in the majority of the indicators. Different from the cities of level I, these cities are located in the hinterlands and have poor natural resources, population emigration, traffic problems and fierce competition within the agglomeration. In this context, it can be predicted that it was a great challenge for local governments to improve their level of sustainability.

Based on the results from single indicators analysis above (Figure 2), Shanghai, Hangzhou, Nanjing, Suzhou and Wuxi had superior performance but the overall sustainable development level was still constrained by some obstacles such as air pollution, the growth of construction land and per capita public green space. At the same time, Tongling, Xuancheng, Anqing, Chizhou, Zhangzhou, Maanshan, Taizhou, Zhoushan and Shanghai were all subject to imbalances of certain indicators. Among them, Shanghai had mainly problems with air pollution, sewage discharge, waste discharge and per capita green space. The more prominent barriers in Zhoushan were the increase of commercial housing area, public transportation and the number of registered households. Other cities encountered challenges that typically arise in the early stage of urbanization such as decreases in urban green space, air quality and the number of registered households. 

### 4.3. Spatial and Temporal Characteristics of Urban Sustainability across the YRD Urban Agglomeration

#### 4.3.1. Spatiotemporal Patterns of Urban Sustainability

As shown in Figure 4, the growth rate of synthesis indicators for most cities in the YRD from 2007 to 2016 increased year by year. Some rules could be found in spatial exploration—the performance was mainly high in the coastal area and low in the inland; meanwhile, it was high in the southeast and low in the northwest. As for the change rate, the southwest exhibited a slowing rate while the south had an increasing rate. Furthermore, when it came to the provincial capital cities, the increase was significant. The coastal areas were represented by Zhoushan, Nantong, Shanghai and Taizhou, while the inland areas were more typical of Xuancheng, Anqing, Maanshan and Tongling. The value of synthesis indicators of Shanghai and most cities of Zhejiang Province were relatively desirable. Referring to Jiangsu and Anhui provinces, the sustainable development of urban development in the northwest corner of the YRD, especially in Yancheng and Zhangzhou, seemed to be weak over the years. From the viewpoint of spatial dimension, these situations in the urban-related SDG-11 index for those cites were largely driven by urban location, resource characteristics, pillar industries and level of economic development. Since the overall industrial transformation of the YRD mainly focused on a set of key fields as the creation and development of equipment manufacturing, information technology, bio-pharmaceuticals, automobiles and new materials, the development speed of cities in the northwest was comparatively slower than that in the coastal areas, due to their spatial location and stages of development [19,35,36].

#### 4.3.2. Spatiotemporal Clustering Analysis

Through the global spatial autocorrelation analysis of the spatial autocorrelation coefficient Moran’s I from 2007 to 2016 (Table 2, Figure 5), it was found that the overall Moran’s I index of the YRD urban agglomeration was low and the improvement of the sustainable development capacity had not formed an aggregative effect yet. Under these conditions, most cities took their own routes in terms of development and the radiation capacity of urban developed did not easily benefit neighboring cities. In the past 10years, inland cities such as Yangzhou, Chizhou, Hefei and Maanshan had high-low, low-low clusters, which indicated that the development of those regions had a negative impact on the sustainable development of surrounding cities. The causes of negative impact could be a city area, industrial developing pattern, construction of transportation network and policy-oriented basis. It is worth noting that high-high clustering only appeared in Hangzhou and Jinhua in 2010. The rise of these two cities led to the development of the surrounding cities. Large cities that took the lead in development maintained this lead over the subsequent years, while small-scale and fringe small cities emerged as new hot spots and developed quickly. However, due to their early stage of development, their advantage was not so significant and stable, as the so-called integrated development had not yet been formed. Obviously, the sustainable development of cities was influenced by the combined effects of natural, economic, cultural and administrative space and was not dominated by any single aspect. Whether the city was promoted by the higher-level government from top to bottom or the regional cooperation was explored by the local government from the bottom up was restricted by the national political system. The marginal and secondary areas engaged in urban agglomeration found difficulties in the integration process [37,38].

## 5. Conclusions

This study introduced the FPPSI method to assess both the single and synthesis indicators for sustainable development of the YRD agglomeration in 2006–2016. The key factors and obstacles of urban sustainable development were identified from the viewpoint of urban agglomeration integration in order to assist in problem diagnosis and policy guidance for the central government. The results implied that:

(1) The FPPSI method is an objective and quantitative method to evaluate the sustainability level and understand the overall effect of policies on SDGs, especially over time. It could not only evaluate the overall level but also analyzed the impact of single indicators by sorting out the superior and inferior indicators. This method has advantages such as ease of calculation and the ability solve multiple indicator evaluation problems, improving the comparability of the valuation scores in different years and grading the comprehensive level of sustainable urbanization.

(2) During the studied years, the synthesis indicators for most cities in the YRD were significantly relieved but each city still showed a unique trend due to their different positions and functions. Meanwhile, there were also some common problems to be solved urgently in the YRD agglomeration. Judging from the carrying capacity of each subsystem, the sustainable development level of small and medium-sized cities was still dominated by the traditional strong sustainability indicators. The sustainable levels of large and mega cities were affected by weak sustainable indicators. 

(3) The spatial distribution and structure of sustainable development in the YRD agglomeration showed difference and imbalance among the cities. The inter annual standard deviations of Shanghai, Yancheng, Xuancheng, Huzhou and Taizhou seemed to increase over the years, which indicated the unbalanced development and instability among the indicators. It is necessary to pay attention to the strong spillover effects of developed cities in the YRD, such as Shanghai, Hangzhou and Ningbo. Therefore, while rationally developing the YRD, regional developing patterns need to be improved, especially in terms of the innovation of internal and external systems in the northern cities of the YRD. This can be achieved by strengthening various forms of integrated cooperation and coordinated development among regions [39,40]. 

## 6. Implications

Judging from the comprehensive evaluation of typical cities, the YRD has realized the initial sustainable development of resource-based cities but the degree and stage of development among those cities vary. The low score indicators should be seen as a wake-up call; those cities should take further action to promote the concept of green and sustainable development to achieve so-called sustainable transition and development in the process of the YRD integration. It is worth noting that new actions in relation to one indicator are very likely to affect scores for other indicators, how they relate to big issues need further study to confirm, such as climate change, loss of biodiversity, poor air quality, poverty and so forth. In the process of the integrated development of the YRD, it is necessary to differentiate between cities according to their background and level, especially to overcome the restrictions and obstruction factors for each city.

The rising trend in the coastal areas is obvious and the sustainability level of large cities is gradually developing as well. However, the developing pressure is proportional to the size of the city and paying scant attention can bring about serious impacts and cause “urban mass.” The less developed cities from inland areas have just set out on the way towards “positive changes;” meanwhile, the pressure to achieve sustainable development has gradually increased. The main reasons for this situation might be the protection of various localities and the existence of competition among neighboring cities. Some regions have higher profits, while other regions have sacrificed their own interests. Therefore, cities call for closer cooperation and more effective policies to strengthen inter-cooperation and to build their capacity to realize sustainable development. In the future, it will be necessary to establish a unified benefit sharing and interest compensation mechanism within the YRD [18,41,42].Only by constructing a scientific compensation mechanism can we improve the advancement of regional functional subdivisions, as well as ensure the smooth connection between traffic lines, resource utilization and ecological environment protection. Assisting small and medium-sized cities on the way to becoming sustainable cities can also relieve the pressure of mega cities. 

## 7. Limitations

We recognize that the FPPSI method has both practical and operational significance in the assessment of the sustainable development level for cities and agglomerations. However, as a result of the spatial differences of observed indicators, the weight of the single indicators needs to be taken into consideration and adjusted appropriately based on the consultation of experienced experts in future research. This would make the results more reasonable and strengthen the practical guiding significance for regional sustainable development. Due to data gaps in the SDGs, more indicators are needed in order to be able to display all different aspects of sustainability. Since the traditional statistical data is short in real-time performance and lacks sufficient spatial completeness, new perspectives and pathways for realizing real-time dynamics, spatial fineness and the quantitative evaluation of sustainability ought to be further explored and addressed. 

## Figures and Tables

**Figure 1 ijerph-16-02288-f001:**
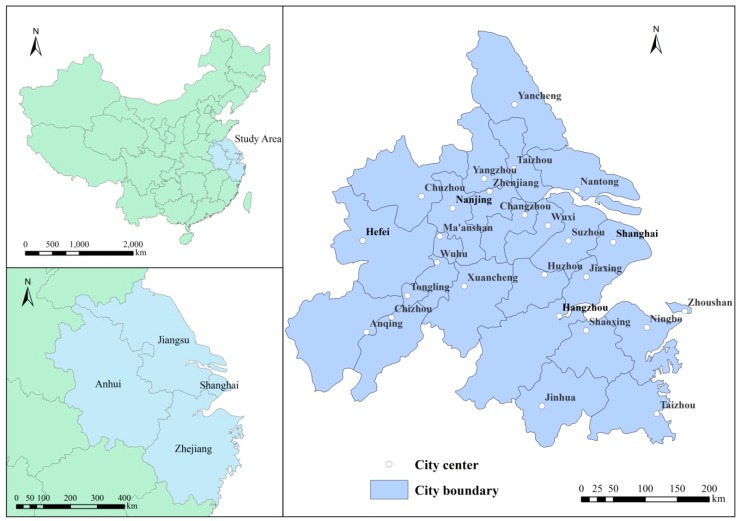
Study area.

**Figure 2 ijerph-16-02288-f002:**
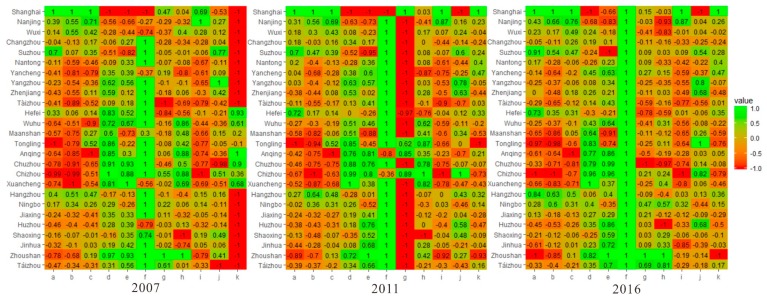
Heatmap of observed urban sustainability index. If a city has an urban-related sustainable development goal (SDG)-11 index above or below a given threshold, the index values are color-coded such that dark red reflects that the observed indicator that is much lower than the zero limit and light green indicates that the observed indicator is much higher than the zero limit. a: commercial housing sales area; b: buses; c: urbanization rate; d: urban sewage discharge; e: solid waste discharge; f: domestic garbage disposal rate; g: PM_2.5_ annual average concentration; h: days of good air quality; i: urban green space area; j: per capita park and green space area; k: household registered population.

**Figure 3 ijerph-16-02288-f003:**
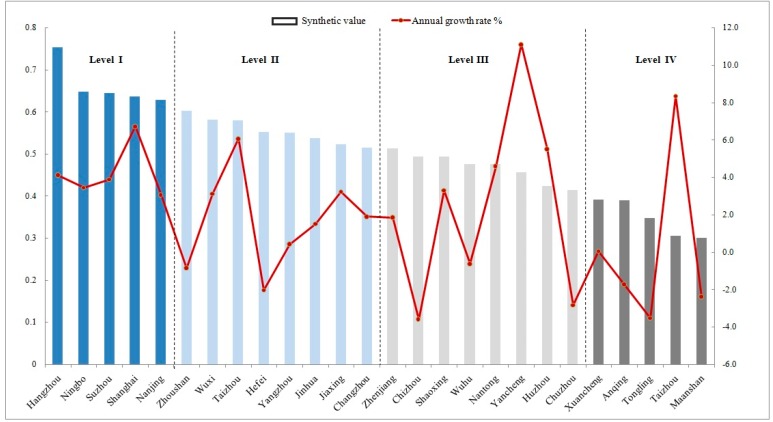
Synthetic indicator for sustainability of 26 cities in the Yangtze River Delta (YRD).

**Figure 4 ijerph-16-02288-f004:**
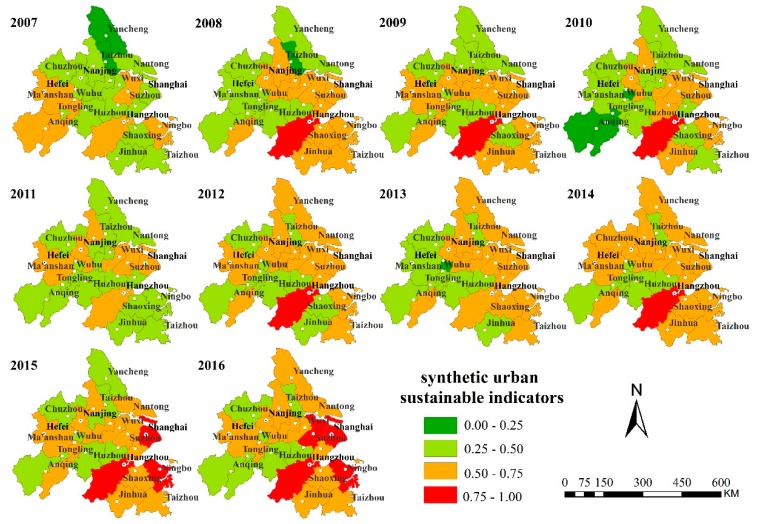
index values of Sustainable Development Goal (SDG)-11-related indicators of the YRD.

**Figure 5 ijerph-16-02288-f005:**
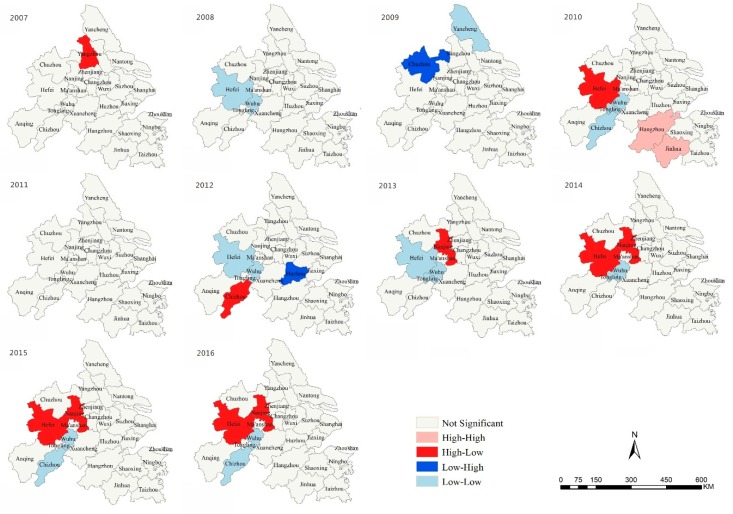
Hotspot areas of urban sustainability in the Yangtze River Delta (YRD).

**Table 1 ijerph-16-02288-t001:** Indicators for urban sustainability assessment.

Indicators	Association Factors	Unit	Indicator Attribute
Housing	(a) Commercial housing sales area (+)	10,000 m^2^	W
Traffic	(b) Number of buses (+)	million	W
Land use and participatory planning	(c)Urbanization rate (+)	%	W
Environmental impact	(d)Urban sewage discharge (−)	10,000 tons	S
	(e)Solid waste discharge (−)	10,000 tons	S
	(f)Domestic garbage disposal rate (+)	%	S
	(g)PM_2.5_ annual average concentration (−)	μg/m^3^	S
	(h)Days of good air quality (+)	day	S
Public space	(i)Urban green space area (+)	hm²	W
	(j)Per capita park and green space area (+)	km	W
Relationship between urban and rural areas	(k) Household registered population (+)	10,000	W

Note: +(−)indicates that the greater(lower)the indicator, the better the level for sustainability; W: Weak sustainability, S: Strong sustainability.

**Table 2 ijerph-16-02288-t002:** Global Moran’s I of sustainable development level in the YRD.

Year	2007	2008	2009	2010	2011	2012	2013	2014	2015	2016
Moran’s I	0.098	0.115	−0.154	0.355	−0.054	0.101	0.156	0.115	0.076	0.184

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
