# Peer review of "An Assessment of Chinese Pathways to Implement the UN Sustainable Development Goal-11 (SDG-11)—A Case Study of the Yangtze River Delta Urban Agglomeration"

_ijerph, 2019, doi:10.3390/ijerph16132288_

Round 1
Reviewer 1 Report
The paper presents a method to measure quantitatively and synthetically the SDG-11 in 26 cities in the Yangtze Delta River. It is an interesting paper due to the highly topical issue and the relatively simple approach of the method. Overall, it is well written and well structured. However, some improvements are required in terms of clarifying specific gaps and issues in the introduction section and in order to sit some research decisions on a stronger basis. Here are some suggestions:
- The first paragraph of section 6 is very clear. It should be taken up in the first paragraph of the introduction. This way, the last sentence (Despite the broad literature on issues…) would be even more supported.
- Line 60: add (Sustainable cities and communities) after SDG-11
- Clarify sentence 89 to 92
- Line 131: Authors should explain what has guided the selection of indicators? Why the number is limited (or extended) to 11? According to the reviewer, it is a lot related to the specific area of YRD, which makes this research interesting while leaving the method adaptable to other cities…but it is not said in the paper.
- Line 131-132: Authors should explain more thoroughly the theory behind weak and strong indicators (look for other references) and its impact on the method.
- Table 1: It would be clearer to add a code for each indicator (such as letters) and refer to it in the text, especially in the results section.
- Indicators Housing and relationship between urban and rural areas are not obvious to understand. What is really measured and what is the impact on sustainability.
- Line 174: The structure of urban development tended to be reasonable…please clarify what it means.
- Line 225: Wuxi had superior indicators…do the authors mean superior performance?
- Figure 3 needs a much more developed legend.
- Section 4.2 is a bit difficult to read and could be improved by including stronger links with figure 3
- Line 310: Authors say that the method can evaluate the sustainability level. As a reader, it is possible to understand that it measures the overall sustainability in a general way. Authors should pay attention to be more specific to the SDG-11. Indeed, the method was not tested for all SDG. This can also be found in other parts of the paper.
- Line 350: “ a slack lack of attention” what does it means precisely? Please change the formulation to something more accurate to the development of cities.
- Section 7 is interesting and some remarks could be included in section 3.1 to support and explain the choice of indicators.
Author Response
Thank you very much for the insightful comments and giving us the opportunity to revise the manuscript. The manuscript has been revised according to the comments,and point-by-point response has also been attached below for your reference.
Response to Reviewers 1#
Comments and Suggestions for Authors
The paper presents a method to measure quantitatively and synthetically the SDG-11 in 26 cities in the Yangtze Delta River. It is an interesting paper due to the highly topical issue and the relatively simple approach of the method. Overall, it is well written and well structured. However, some improvements are required in terms of clarifying specific gaps and issues in the introduction section and in order to sit some research decisions on a stronger basis. Here are some suggestions:
The first paragraph of section 6 is very clear. It should be taken up in the first paragraph of the introduction. This way, the last sentence (Despite the broad literature on issues…) would be even more supported.
[Response]: Thanks. We have reconsidered and reorganized it. Please see Line 54-60.
- Line 60: add (Sustainable cities and communities) after SDG-11
[Response]: Thanks for your valuable suggestions. We have revised. Please check.
- Clarify sentence 89 to 92
[Response]: We have split the whole sentence into two short ones in order to make it more concise. Please check.
Line 131: Authors should explain what has guided the selection of indicators? Why the number is limited (or extended) to 11? According to the reviewer, it is a lot related to the specific area of YRD, which makes this research interesting while leaving the method adaptable to other cities…but it is not said in the paper.
[Response]: Thanks for your valuable comments. Here, we pay more attention to explain our selection of indicators in details. For example, we added more content at the beginning of section 3.1 ; By adding so much content, the authors hope that the addition would meet the requirement..
Line 131-132: Authors should explain more thoroughly the theory behind weak and strong indicators (look for other references) and its impact on the method.
[Response]: Thank you for your kindly remind. We have added two short paragraphs to explain how we applied the theory in this paper. For example," With respect to Urban sustainability science……" and " The final list of 11 indicators considered……". Please see Line 138-151.
Table 1: It would be clearer to add a code for each indicator (such as letters) and refer to it in the text, especially in the results section.
[Response]: Thanks. We have revised. Please check.
Indicators Housing and relationship between urban and rural areas are not obvious to understand. What is really measured and what is the impact on sustainability.
[Response]: Thanks. We believe that in China household registered population of urban can measure population mobility between urban and rural areas. All those observed indicators have referred to the introduction of SDGs, which is quite authoritative. And the source of data is the statistics yearbook with synthetic calculation of high credibility.
Line 174: The structure of urban development tended to be reasonable…please clarify what it means.
[Response]: Thanks. We have reconsidered and revised it.
Line 225: Wuxi had superior indicators…do the authors mean superior performance?
[Response]: Thanks for your valuable comments. We are sorry that this is a mistake of wording. Higher indicators mean superior performances of Wuxi in urban sustainability. We have revised.
Figure 3 needs a much more developed legend.
[Response]: Thank you for your kindly reminding and we have revised, please check.
- Section 4.2 is a bit difficult to read and could be improved by including stronger links with figure 3
[Response]: Thank you for your kindly reminding and we have reorganized it, pleases see Line 239-278.
Line 310: Authors say that the method can evaluate the sustainability level. As a reader, it is possible to understand that it measures the overall sustainability in a general way. Authors should pay attention to be more specific to the SDG-11. Indeed, the method was not tested for all SDG. This can also be found in other parts of the paper.
[Response]: We are more concerned about urban sustainable development, and SDG-11 is a key index system about the urban sustainability. Thanks for your valuable suggestions. The evaluation based on all SDGs index will be studied in our future research.
- Line 350: “ a slack lack of attention” what does it means precisely? Please change the formulation to something more accurate to the development of cities.
[Response]: Thank you for your kindly reminding and we have revised, please see Line 367.
Section 7 is interesting and some remarks could be included in section 3.1 to support and explain the choice of indicators.
[Response]: Thank you for your interest and affirmation. We have reconsidered and revised. Please see Line 147-149 and revised Table 1.
Reviewer 2 Report
As it stands, in my view this paper is not suitable for publication. Standard of English not good enough to allow the reader to understand your text. Limitations section: limitations so great as to reduce the interest in your results. Methods not clearly described. Indicators few in number and their selection is poorly explained. References not formatted correctly. Line 449 a paper is referenced to 2018. I cannot find this paper. I can find a paper published in 2017 with the same title and journal, but with a different first author. All references need to be in standardised format. You make recommendations through the text. Either delete these or bring them together in a conclusion.
Author Response
Thank you very much for the insightful comments and giving us the opportunity to revise the manuscript. The manuscript has been revised according to the comments,and point-by-point response has also been attached below for your reference.
Response to Reviewers 2#
Comments and Suggestions for Authors
As it stands, in my view this paper is not suitable for publication. Standard of English not good enough to allow the reader to understand your text. Limitations section: limitations so great as to reduce the interest in your results.
[Response]: Thank you for your kindly reminding that the language is still need to polish. Our manuscript has been sent to the recommended editing institution from MDPI. The text has been checked for correct use of grammar and common technical terms. We hope that the quality of writing could fulfill the request of this journal.
Methods not clearly described. Indicators few in number and their selection is poorly explained. References not formatted correctly.
[Response]: Thanks for your valuable comments. Here, we pay much attention to explain our selection of indicators more in detail this time. Such as, we added more content at the beginning of section 3.1 ; By adding so much content, the authors hope that the addition would meet the requirement..
Line 449 a paper is referenced to 2018. I cannot find this paper. I can find a paper published in 2017 with the same title and journal, but with a different first author. All references need to be in standardized format.
[Response]: Thank you for your kindly remind. We have checked the reference one by one in the manuscript and also sent our manuscript to the recommended editing institution from MDPI.
You make recommendations through the text. Either delete these or bring them together in a conclusion.
[Response]: Thank you for your kindly reminds. We have reconsidered and revised. We have brought "A low score should be seen as a wake-up call……"in 4.2 to conclusion section.” Please check.
Reviewer 3 Report
Article Title: An Assessment of Chinese Pathways to Implement the UN Sustainable Development Goal-11 (SDG-11)—A Case Study of the Yangtze River Delta Urban Agglomeration
Review Report for International Journal of Environmental Research and Public Health
Recommendation: REJECT – I would NOT recommend this article for publication.
The paper is NOT consistent with MDPI International Journal of Environmental Research and Public Health and does not fit in with the overall journal scope.
Upon reviewing the article topic on “Sustainability” + “Assessment” + “Indicators” and using the authors own Keywords (i.e., urban sustainability assessment; sustainable development goals; full permutation polygon synthetic indicator (FPPSI) method; Yangtze River Delta urban agglomeration), there are ZERO articles even close to the topic being presented.
The authors should re-submit to a more appropriate journal, e.g., MDPI Sustainability, similar keyword search results show over a 150 articles within the scope of the paper.
After reviewing the article in detail, the SGD-11 is to make cities and human settlements inclusive, safe, resilient, and sustainable. The authors use the full permutation polygon synthetic indicator (FPPSI) method to conduct their research. The paper integrates Weak and Strong Sustainability concepts and mapping results which are very well done and merit a further review as urban sustainable indicators are very important for the region (in China) as well as for the greater scientific community. The English is a little bit awkward at times with occasional grammatical errors; I would recommend an English editor revise the manuscript.
Author Response
Thank you very much for the insightful comments and giving us the opportunity to revise the manuscript. The manuscript has been revised according to the comments,and point-by-point response has also been attached below for your reference.
Response to Reviewers 3#
Recommendation: REJECT – I would NOT recommend this article for publication.
The paper is NOT consistent with MDPI International Journal of Environmental Research and Public Health and does not fit in with the overall journal scope.
Upon reviewing the article topic on “Sustainability” + “Assessment” + “Indicators” and using the authors own Keywords (i.e., urban sustainability assessment; sustainable development goals; full permutation polygon synthetic indicator (FPPSI) method; Yangtze River Delta urban agglomeration), there are ZERO articles even close to the topic being presented.
The authors should re-submit to a more appropriate journal, e.g., MDPI Sustainability, similar keyword search results show over a 150 articles within the scope of the paper.
After reviewing the article in detail, the SGD-11 is to make cities and human settlements inclusive, safe, resilient, and sustainable. The authors use the full permutation polygon synthetic indicator (FPPSI) method to conduct their research. The paper integrates Weak and Strong Sustainability concepts and mapping results which are very well done and merit a further review as urban sustainable indicators are very important for the region (in China) as well as for the greater scientific community.
[Response]: We believe that our research is within the range of environment domain. In this paper we have evaluated the sustainability for cities, which is an important application in environmental research. IJERPH has published many papers cover quite a lot of topics in sustainable research recent years, such as” SDGs-3”, “Sustainable Food System”,” Sustainable Employability”, etc. The authors hope the elaborate revision would satisfy the requirements of the reviewer and finally accept the publishing, which can help us to find more collaborators on the world basis to further our future research.
The English is a little bit awkward at times with occasional grammatical errors; I would recommend an English editor revise the manuscript.
[Response]: Thank you for your kindly reminding that the language is still need to polish. We have checked the words one by one in the manuscript and replaced some informal phraseology. And also our manuscript has been sent to the recommended editing institution from MDPI .The text has been checked for correct use of grammar and common technical terms We hope that the quality of writing could fulfill the request of this journal.
Reviewer 4 Report
Dear authors, congrats for the paper and the research work. The research design is very good as well the spatial significance.
My only concern upon scientific robustness and therefore, the results and conclusions significance lies on the chosen method: FPPSI. We can, based on the literature and even empirical knowledge, chose an specific method to measure and analyse our data. However, to make prove of evidence of the suitability of the method to our study, eg. our data and quality of our data, I'm normally like to do, and see, some calibration analysis, methods comparison and or some parameterization. There are other methods that can reflect the integrative system, such as machine learning and bayesian analysis.
Another, although minor, issue I've with this paper it's relate with one of the objective. On line 104 of your paper, you said that "This study also tries to form a paradigm ...". So, my question is: what are the current paradigm and why it's needed to be changed? and please do consider, that a paradigm means a structural change, a revolution, and not just an improvement or an adjust of the model in use.
Author Response
Thank you very much for the insightful comments and giving us the opportunity to revise the manuscript. The manuscript has been revised according to the comments,and point-by-point response has also been attached below for your reference.
Comments and Suggestions for Authors
Dear authors, congrats for the paper and the research work. The research design is very good as well the spatial significance.
My only concern upon scientific robustness and therefore, the results and conclusions significance lie on the chosen method: FPPSI. We can, based on the literature and even empirical knowledge, chose a specific method to measure and analysis our data. However, to make prove of evidence of the suitability of the method to our study, e.g. our data and quality of our data, I'm normally like to do, and see, some calibration analysis, methods comparison and or some parameterization. There are other methods that can reflect the integrative system, such as machine learning and bayesian analysis.
[Response]: Both machine learning and Bayesian analysis are far superior methods which could have a wide range of applications in such evaluation. The method for the further investigation of SDG assessment will be studied in our future research.
Since the sustainability criteria are often selected from different subsystems and there are many criteria, FPPSI is relatively a classic method for data standardization and indicator synthesis to date, which could enable the sensitive identification of cities with upper and lower values for each indicator, and highlights hot spots and cold spots visibly.
On line 104 of your paper, you said that "This study also tries to form a paradigm...". So, my question is: what are the current paradigm and why it's needed to be changed? And please do consider, that a paradigm means a structural change, a revolution, and not just an improvement or an adjust of the model in use.
[Response]: Thanks for your valuable comments and encouraging words. We are sorry for not using the proper word. We have revised by use the word "scientifically" instead of "paradigm".
The authors hope the elaborate revision would satisfy the requirements of the reviewer and the journal, which can help us to find more collaborators on the world basis to further our future research.
Round 2
Reviewer 2 Report
The ms has been improved. As the authors state, starting with a SDG as a basis for setting the goal and ambition of urban sustainable development created problems for them. Fig 3 still has a trend line, but a red dot on each column would be more appropriate. Line 194 the phrase 'tend to be reasonable' needs explanation: people differ in what they regard as reasonable. Line 275 'contradiction' is the wrong term, but I'm not clear on what the right term should be. Line 372 no analysis provided to support the assertion that the method is 'superior'. Superior to what, and why? New actions in relation to one indicator are very likely to affect scores for other indicators: how is this to be tackled in quantifying change over time? Best practice waste management regulation might be expected to reduce profitability. The degree of relevance of findings to the problem of making cities more sustainable is not clear to me. I'm not sure how they relate to big issues such as climate change, loss of biodiversity, poor air quality, etc. The level of ambition in terms of actions to be taken seems very rather low, again perhaps because of the inherent vagueness of the SDG. The Limitations section is better, and points to various crucial issues which, however, need to be better reflected in the preceding text. Still little explanation of the method for selection of indicators was undertaken, other than relevance to the SDG. Recommendations made in relation to what new and more established cities might do to enhance their sustainability seem to me too incomplete and generalised to support prioritisation of corrective actions actions, and should be deleted.
The ms presents a first and necessarily limited demonstration of new methods, and visual ways of representing spatial data. As it stands, it is not clear to me that application of these new methods has provided novel and useful insights. However, the results might be seen as indicative of what could be achieved were the methods developed further, that is, the issues identified in Limitations are more fully addressed. If framed in this way, a shorter version of the ms might justify publication.
Author Response
Thank you very much for insightful comments and suggestions, which are really helpful for revising and improving our ms. We have tried very hard to address all marks raised and hope the revised ms will meet with approval. The responds to your comments has been attached.

Reviewer 3 Report
The paper is very well written and merits publication. I do not believe, however, the topic is within the scope of the journal. I therefore would advise the authors to submit the manuscript to a different, more appropriate journal (e.g., MDPI Sustainability).
